# SOCA-PRNet: Spatially Oriented Attention-Infused Structured-Feature-Enabled PoseResNet for 2D Human Pose Estimation

**DOI:** 10.3390/s24010110

**Published:** 2023-12-25

**Authors:** Ali Zakir, Sartaj Ahmed Salman, Hiroki Takahashi

**Affiliations:** 1Department of Informatics, Graduate School of Informatics and Engineering, The University of Electro-Communications, Tokyo 182-8585, Japan; s2140019@edu.cc.uec.ac.jp (S.A.S.); rocky@inf.uec.ac.jp (H.T.); 2Artificial Intelligence Exploration Research Center/Meta-Networking Research Center, The University of Electro-Communications, Tokyo 182-8585, Japan

**Keywords:** 2D human pose estimation, CV, SOCA-PRNet, Global Context Blocks

## Abstract

In the recent era, 2D human pose estimation (HPE) has become an integral part of advanced computer vision (CV) applications, particularly in understanding human behaviors. Despite challenges such as occlusion, unfavorable lighting, and motion blur, advancements in deep learning have significantly enhanced the performance of 2D HPE by enabling automatic feature learning from data and improving model generalization. Given the crucial role of 2D HPE in accurately identifying and classifying human body joints, optimization is imperative. In response, we introduce the Spatially Oriented Attention-Infused Structured-Feature-enabled PoseResNet (SOCA-PRNet) for enhanced 2D HPE. This model incorporates a novel element, Spatially Oriented Attention (SOCA), designed to enhance accuracy without significantly increasing the parameter count. Leveraging the strength of ResNet34 and integrating Global Context Blocks (GCBs), SOCA-PRNet precisely captures detailed human poses. Empirical evaluations demonstrate that our model outperforms existing state-of-the-art approaches, achieving a Percentage of Correct Keypoints at 0.5 (PCKh@0.5) of 90.877 at a 50% threshold and a Mean Precision (Mean@0.1) score of 41.137. These results underscore the potential of SOCA-PRNet in real-world applications such as robotics, gaming, and human–computer interaction, where precise and efficient 2D HPE is paramount.

## 1. Introduction

The widespread use and improvement in computer vision (CV) technology in various everyday settings, such as smartphones, digital cameras, and surveillance systems, generate a constant stream of image and video data. Extracting information about human activities from this data is of great importance. Central to these interaction mechanisms is HPE. HPE focuses on identifying and categorizing various joints in the human body. It captures each joint’s coordinates, such as arms, head, and torso—often termed keypoints, to delineate a person’s posture. Over recent decades, the automated interpretation of HPE has become a significant research interest within the field of CV. It forms the foundation for numerous complex CV tasks. It provides a base for predicting 3D HPE, identifying human actions and motion prediction, parsing human body components, and retargeting human movements. Additionally, 2D HPE offers substantial support across applications, from understanding human dynamics, monitoring crowd anomalies or riots, spotting instances of violence, detecting unusual behaviors, and enhancing human–computer interaction (HCI) to aiding autonomous vehicle advancements [1]. The complexity of 2D HPE stems from various factors, occluded keypoints, challenging lighting and background conditions, motion blur, and the intimidating task of implementing the model in real-time due to its vast number of parameters [2].

In the initial phases of research in 2D HPE, the field predominantly relied on traditional methods such as probabilistic graphical models [3,4]. These approaches were characterized by a considerable dependence on manually designed features incorporated into models. While effective to an extent, this reliance on handcrafted features often posed significant limitations, restricting the models’ capacity for broader generalization and optimal performance. The intricate nature of human poses, varying across diverse contexts and environments, posed challenges that these traditional methods struggled to consistently address.

As the field evolved, a paradigm shift occurred with the advent of deep learning techniques. This marked a substantial transformation in the approach to 2D HPE. Deep learning, diverging from the constraints of manual feature engineering, brought the capability of automatically extracting relevant features and learning from data. This shift was particularly catalyzed by the advancements in convolutional neural networks (CNNs). CNNs’ ability to process complex image data effectively and their versatility in learning feature hierarchies propelled 2D HPE into a new era. The success of CNNs and their applications in pose estimation underscored the potential of deep learning, paving the way for the development and incorporation of various sophisticated deep learning strategies that built on the foundational achievements of CNNs [5].

With this backdrop, the primary objective of our paper is to further enhance the prediction accuracy of 2D HPE methods while optimizing efficiency through a reduced parameter set. We recognize the challenges posed by large deep learning models, particularly when deployed in real-time or resource-constrained settings. Such models, while powerful, can be computationally demanding, memory-intensive, and may require specialized hardware. Additionally, their complexity often risks overfitting, where performance on training data does not translate to unseen data. Addressing these concerns, our research aims to strike a balance between accuracy and efficiency, creating versatile and cost-effective models suitable for a range of applications, from edge computing devices to large-scale cloud infrastructures. This endeavor leads us to propose SOCA-PRNet, a framework that epitomizes this balance by integrating advanced features within a streamlined architecture.

Our research led us to the simple baseline network [6], which has demonstrated superior performance compared to other top-down methodologies. Its streamlined and efficient architecture positions it as a prime foundation for further advancement in 2D HPE. Building on this foundation, we introduce SOCA-PRNet. This framework is characterized by integrating a Spatially Oriented Attention-Infused Structured Features module, with a modified version of ResNet serving as its primary feature extractor [7,8]. Within this ResNet adaptation, we have omitted the average pooling and the last fully connected layers, emphasizing convolutional layers. To further simplify the model and decrease its complexity, we have employed ResNet34 over the more elaborate variants such as ResNet50, 101, or 152, all of which possess a larger parameter count. We have added two deconvolution layers designed to enhance visual processing capabilities and mitigate quantization distortions from large output stride sizes. While it is understood that a smaller network size might impact the model’s accuracy due to the trade-off between precision and parameter quantity, we have addressed this by another significant inclusion is the integration of Global Context Blocks (GCBs) [9], which aims to expand the performance of both the downsampling and upsampling modules. Furthermore, our innovative SOCA module merges and amplifies feature representations through spatial attention, channeling these refined features to the upsampler layers, thereby bypassing traditional skip connections [10,11]. This methodology fosters hierarchical representations with enhanced spatial awareness, adeptly capturing complex details. These modifications and attributes are designed to offer a detailed, context-rich representation of data, ensuring the model’s stability.

The threefold contribution of the SOCA-PRNet model can be summarized as follows:We introduced the SOCA-PRNet, deliberately choosing ResNet34 over more intricate models to streamline its structure. This decision promotes efficiency without sacrificing capability. Further enhancements include adding two deconvolution layers, bolstering the model’s visual processing, and addressing the quantization distortion from large output stride sizes. We integrated GCBs into the downsampler and upsampler modules to endow the model with robust global context features.Central to our design is the SOCA module. It merges features derived from various downsampler layers. These collective features undergo refinement via a spatial attention mechanism and are subsequently channeled to the appropriate upsampler layers. The outcome is a generation of hierarchical representations with enhanced spatial awareness, adeptly capturing intricate pose details.To evaluate the merits of our proposed model, we subjected it to rigorous testing on the MPII dataset. Both quantitative and qualitative assessments revealed that SOCA-PRNet outperforms existing 2D human pose estimation techniques in terms of accuracy while maintaining a more favorable computational cost.

This article follows a structured approach, with several sections. Section 2 presents an overview of prior research conducted in the same field. Section 3 elaborates on the comprehensive methodology of our proposed SOCA-PRNet. Section 4 covers pertinent information regarding the experimental setup and implementation details. An analysis of both qualitative and quantitative results is exhibited in Section 5. Section 6 offers an in-depth analysis of our results.The final section, Section 7, draws conclusions and lays out plans for future exploration.

## 2. Related Works

Deep learning approaches are utilized in designing network architectures for 2D HPE to extract robust features that span from low to high levels. These approaches are typically categorized into two frameworks: the top-down and bottom-up frameworks. The method of the top-down paradigm involves a sequential process where the initial step is to identify the human bounding boxes in an image, followed by executing the single HPE for every identified box. This type of approach is not a suitable method for managing large crowds as the computational time for the second step increases in association with the number of individuals present [1,8]. A. Toshev et al. [12] have made a pioneering contribution to the field of HPE by introducing a CNN for the first time.

They leveraged the CNN’s robust fitting capability to regress the coordinates of human joints implements a cascading structure to refine the outcomes continuously, though the model tends to overfit because the weights of the fully connected layer depend on the distribution of the training dataset. The convolutional pose machine (CPM) [13] and stacked hourglass networks [10] solved this issue by predicting heatmaps of 2D joint locations. Two main object detection techniques exist in 2D HPE: the RCNN [14] series and the SSD series [15]. The RCNN series employs a complicated network structure that achieves high accuracy and introduces the Mask-RCNN approach, which builds upon the faster RCNN architecture [14] by incorporating keypoint prediction. As a result, this method achieves excellent results in HPE, demonstrating strong competitiveness in this domain. Conversely, the SSD series offers an average compromise between precision and Y. Chen et al. [16] present the concept of a cascaded pyramid network (CPN) that uses GlobalNet to identify simple keypoints and Refine-Net to handle more challenging keypoints. To be more precise, Refine-Net includes multiple standard convolutional layers that merge feature representations from all levels of GlobalNet.

The process of bottom-up methods starts with detecting keypoints for every human instance present in an image. Subsequently, the keypoints of the same individual are joined to form skeletons of multiple instances. This grouping optimization problem is crucial in determining the outcome of the bottom-up approach. Some representative methods utilize this approach, and they are [5,17]. Open-Pose, as described in [5], utilized two branches—one of which employed a CNN to predict all keypoints based on heatmaps, and the other used a CNN to acquire part affinity fields. The part affinity fields represent 2D direction vectors, and they serve as a confidence metric to determine if the keypoints are associated with the same person. Ultimately, both branches are merged to generate the concluding prediction. The approach known as associative embedding [11], derived from hourglass networks [10], is end-to-end trainable. The source detected and accumulated keypoints in one step without requiring two separate processes.

Implementing bottom-up approaches can be challenging due to the difficulty of combining information from multiple scales and grouping features together. Even with the introduction of effective grouping procedures, these methods still struggle to contest top-down strategies for pose estimation. In recent times, the majority of cutting-edge outcomes have been achieved through top-down methodologies. Our research traced the top-down approach and developed a successful 2D HPE model. This addresses the issue of top-down approaches by modifying a baseline network with Spatially Oriented Attention-Infused Structured Features. We utilized a simpler ResNet34 model and removed specific layers to reduce complexity. We then added deconvolution layers and GCB to improve visual processing and global context features. The proposed SOCA module combines and enhances feature representations from various layers, enabling better capture of finer details through hierarchical representations with spatial awareness.

## 3. Proposed SOCA-PRNet

We introduce SOCA-PRNet, a novel framework in the field of 2D HPE, distinguished by its integration of a SOCA module with a modified ResNet architecture. This framework is designed to address the intricate requirements of pose estimation by enhancing feature representation and spatial awareness. Our approach begins with the primary objective of 2D HPE; given an RGB image or a video frame labeled as *I*, the goal is to identify the posture. The pose P of any individual is represented in this visual content. This posture, expressed as P, is characterized by a set of *N* specific keypoints. Each keypoint is denoted by a two-dimensional coordinate $\left(x_{n},y_{n}\right)$. The number of keypoints, *N*, can vary based on the dataset used for training a model. Thus, our objective is to pinpoint the pose $P={\left\{P_{i}\right\}}_{i=1}^{N}$ for every *k* individual within the input. Algorithm 1, while general, represents the fundamental process of pose estimation in the field of 2D HPE. It serves as a baseline framework from which the innovations of SOCA-PRNet are developed. The algorithm outlines the standard procedure of initializing the posture set, detecting individuals in the image, identifying keypoints, and compiling these into a posture representation.
**Algorithm 1:** Foundational process of 2D human pose estimation (HPE)
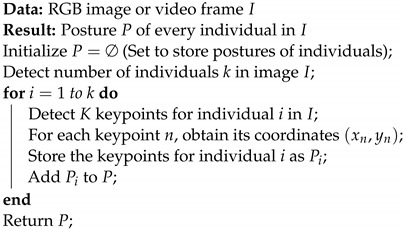


In developing SOCA-PRNet, we adapted the ResNet architecture, emphasizing convolutional operations and reducing complexity. Specifically, we chose ResNet34 for its balance of efficiency and performance and added two deconvolution layers to enhance visual processing. We also integrated GCBs to improve both downsampling and upsampling modules and introduced the SOCA module, a key innovation that merges and amplifies feature representations through spatial attention. This module directs refined features to the upsampler layers, effectively bypassing traditional skip connections. These modifications aim to provide detailed, context-rich data representations, ensuring both stability and accuracy in pose estimation.

Building on the foundational process outlined in Algorithm 1, the SOCA-PRNet introduces specific enhancements. These include the integration of the SOCA module and the ResNet architecture’s adaptation, which collectively enhance the accuracy and efficiency of pose estimation. This advanced approach, leveraging novel techniques for feature representation and spatial awareness, marks a significant evolution from the general framework of 2D HPE.

Figure 1 presents the detailed structure of SOCA-PRNet, clearly explaining it from the simple baseline network as shown in Figure 2 [6], upon which our research builds. The figure is designed to distinctly show the architectural changes and the inclusion of novel components unique to SOCA-PRNet. Key differences are highlighted, such as the replacement of certain ResNet layers with GCBs and the addition of the SOCA module. These differences are visually contrasted against the architecture of existing networks, emphasizing the enhancements and optimizations we have incorporated.

In the following subsections, we explore the structure of SOCA-PRNet as depicted in Figure 1. We will provide a comprehensive explanation of each component, from the modified ResNet base through the integration of deconvolution layers and GCBs to the final integration of the SOCA module. This detailed breakdown will clarify the functionality of each element in our model and explain how these components collaboratively contribute to the model’s overall performance, highlighting the advancements our network brings to 2D HPE.

### 3.1. Enhancing Backbone Model with Modified ResNet and Deconvolution Module

The structure of the residual network is commonly utilized for dense labeling tasks. To achieve this, we employed such kind of network structure that slowly decreases the resolution of embeddings to capture extended-range details, which subsequently increases feature maps while recovering spatial resolution. Hourglass and simple baseline networks create smaller output feature maps than their input feature maps, which are then resized using a simple transformation technique that can cause quantization errors. When data processing is biased, prediction errors can occur due to horizontal flipping and how the model processes the output resolution [18]. We incorporated two deconvolution modules into our approach to tackling the above mentioned challenges. These modules were designed to generate a complete output feature map and were integrated within the architecture of the simple baseline network. We opted to use ResNet34, which has fewer parameters than more complex ResNet models like 50, 101, or 152. We modified ResNet [7] by removing the average pooling segment and fully connected part and replacing them with four ResNet blocks after a convolutional and pooling layer. The modifications are visually depicted in Figure 3. The first set of layers in the network, which includes a convolutional layer and a pooling layer, reduces the size of the feature maps by half. As the input passes through each block of the network, additional convolutional layers are used to decrease the feature maps by two strides while simultaneously increasing the number of filters by a factor of two. We added five deconvolutional modules with batch normalization and HardSwish activation, each doubling the feature resolution map until the output matches the input. The fourth and fifth deconvolutional layers have channel sizes of 64 and 32, respectively.

### 3.2. Amplifying Model Performance with GCB

In computer vision, a Global Context Block is a module designed to capture the overall spatial information of an input feature map, aiming to improve object recognition in an image. In convolutional layers, the association among pixels is only considered within a local neighborhood and baseline network. We opted to use ResNet34, which have fewer parameters compared to more complex ResNet; capturing long-range dependencies requires multiple convolution layers. To address this limitation, researchers proposed a non-local operation [19], which employed a self-attention mechanism from [20] to model long range dependencies. Using a global network creates an attention map tailored to each query position, enabling the collection of contextual features that can then be integrated into the features of the corresponding position. GCNet is presented as a highly well organized and operative method for global context modeling [9]. This method employs a query agnostic attention map to generate a contextual representation that can be globally shared and then incorporates it into the features of each query location in the network.

Our proposed method uses GCBs [9] to enhance the spatial information of input feature maps. Specifically, as illustrated by the sky-blue blocks in Figure 1, GCBs are incorporated into each ResNet block as well as the first four blocks of the deconvolution modules. We generate a spatially aware attention heatmap using a 1 × 1 convolution and SoftMax to produce attention weights, which are then used in attention pooling to extract a global context feature. Channel-wise dependencies are obtained using the bottleneck transform technique. Afterward, the resulting global context features are combined with the features of each position in the network, as shown in the following Equation (Equation 1).
(1)$$f_{g}=\sum_{i=1}^{h}\sum_{j=1}^{w}w_{ij}f_{ij}$$
where in Equation (Equation 1), $f_{g}$ represents the global context feature, *h* and *w* are the height and width of the input feature map, $w_{ij}$ is the attention weights at position (i,j), and $f_{ij}$ is the feature vector at the position (i,j).

### 3.3. SOCA Module

The Spatially Oriented Attention-Infused Structured Features (SOCA) module overcomes the limitations of earlier frameworks, such as the simple baseline framework [6], which did not integrate skip connections [10,21]. These connections have proven effective in U-Net and hourglass networks for retaining spatial information at each feature map, allowing for an efficient transfer of spatial information across the network, and leading to improved localization.

In contrast to these earlier approaches, our SOCA module, as depicted in Figure 4, represents a significant advancement. Unlike traditional skip connections that typically rely on direct concatenation or summation of feature maps, SOCA employs a novel approach of combining hierarchical features from various layers. It utilizes spatial attention to selectively enhance features that are critical for pose estimation. This process involves the elementwise multiplication of feature maps from the first four Global Context Blocks, ResNet blocks, and spatially oriented attention feature maps. As a result, SOCA provides a more targeted enhancement of features, emphasizing areas crucial for accurate pose estimation. The design of the SOCA module is specifically tailored to generate more relevant details by focusing on key locations for pose estimation while effectively suppressing less relevant background information. This leads to a significant improvement in feature specificity, which is crucial for pose estimation tasks. The spatially aware attention mechanism of SOCA ensures that the enhanced features are optimally tuned to the demands of pose estimation, contributing to robust and accurate model performance, especially in complex scenes.

Our analysis further highlights the advantages of the SOCA module over traditional skip connections. The method of feature integration used by SOCA, through spatial attention and elementwise multiplication, aligns well with tasks that require high accuracy in localization. This approach offers a more refined and context-aware integration of features compared to the simpler methods used in traditional skip connections. By examining the existing literature on skip connections and spatial attention mechanisms, we underline the improvements that SOCA brings in terms of feature representation and model performance. The enhancements in performance with the integration of the SOCA module are evident in the experimental results and analysis section.attest to its effectiveness. Analyzing these data in various pose estimation scenarios reveals the practical benefits of SOCA over conventional methods.

Our observation indicates that the SOCA module is a more effective feature combination mechanism for 2D HPE models compared to traditional skip connections. SOCA’s focus on spatially oriented feature enhancement is expected to lead to improved accuracy in pose estimation, particularly in complex and varied scenarios.

### 3.4. Heatmap Joint Prediction

Our model employs a sophisticated approach to estimate joint positions by transforming pixel-level predictions into a spatial probability distribution, represented as heatmaps. This transformation is facilitated by a 2D Gaussian function centered on each joint’s true location within the confines of a bounding box. The intensity at each pixel location (x,y) on the heatmap $H{}_{k}\left(x,y\right)$ is computed in Equation (Equation 2):(2)$$H_{k(x,y)}=\exp\left(\frac{-\left[{\left(x-y_{k}\right)}^{2}+{\left(y-y_{k}\right)}^{2}\right]}{2\sigma^{2}}\right)$$

In Equation (Equation 2), $H_{k}$ represents the heatmap for $k^{\mathrm{th}}$ joint where k∈{1,2,…,K}, and (x,y) show the position of the specified pixel in the heatmap. The $k^{\mathrm{th}}$ joints coordinated are denoted by $\left(x_{k},y_{k}\right)$. After several experimental iterations, we found that setting σ to 6 offers an optimal balance, capturing the joint’s essence without excessive spreading.

## 4. Experimental Setup

### 4.1. Dataset

Our experimentation to evaluate the effectiveness of our proposed model utilized the widely recognized MPII (Max Planck Institute for Informatics) Dataset [22]. This expansive dataset encompasses over 25,000 annotated images, representing more than 40,000 individuals, each annotated with 16 unique keypoints. We strategically divided this dataset into two subsets: one for training and the other for testing. For the training phase, we used 28,000 images to develop and fine-tune our model. Subsequently, we reserved a separate set of 11,000 images to evaluate the model’s performance, offering an objective measure of its robustness and accuracy.

### 4.2. Implementation Details

We employed a range of data augmentation techniques to enhance our model’s robustness against scale and spatial rotation variations. We introduced a random horizontal flip to the dataset, diversifying its content and helping to mitigate overfitting. Additionally, we made rotation adjustments, enabling the model to process images tilted between −40 and +40 degrees, which improves its adaptability to varied orientations. To ensure the model effectively recognizes objects of different sizes, we adjusted scaling within a range of 0.7 to 1.3. The model was built using the PyTorch framework. During training, we set a learning rate of $1\times 10^{-5}$, used a batch size of 16, and deployed six workers for parallel data processing. The training process extended over 150 epochs.

### 4.3. Loss Calculation Function

Using the correct loss function is crucial for precise model training. In our methodology, we opted to utilize the Mean Square Error (MSE) loss function, which has been utilized in previous works such as [6,8,23], to evaluate the model’s error. The formula for MSE is presented in Equation (Equation 3).
(3)$$L=\frac{\sum_{k=1}^{K}∥H_{k}-{\hat{H}}_{k}∥}{K}$$

In Equation (Equation 3), ${\hat{H}}_{k}$ represent the estimated heatmap for the $k^{\mathrm{th}}$ joint, whereas $H_{k}$ is the heatmap for the $k^{\mathrm{th}}$ joint k∈{1,2,…,K}. In our implementation, we have provided the option to weight the errors for different joints differently using target-weight. If this option is enabled, the MSE for each joint is computed after multiplying the heatmaps by their respective weights. This allows us to prioritize or deprioritize certain joints based on their importance or reliability in the dataset.

### 4.4. Optimization of Model

The optimization of models can be enhanced with the implementation of a new variant of the Adam optimizer named AdamW. The key difference between Adam and AdamW lies in their approach to weight decay. While Adam intertwines weight decay with its adaptive learning rate—sometimes resulting in suboptimal training dynamics—AdamW distinctly decouples weight decay from learning rate updates. This separation minimizes the interference between the adaptive learning rate and weight decay, fostering more stable and consistent optimization. Consequently, models optimized with AdamW tend to exhibit less overfitting than those using the original Adam, making AdamW a favored choice among many deep learning practitioners. We conducted a series of experiments, the comprehensive details of which are presented in Section 6 and summarized in Table 3.

### 4.5. Activation Functions

In deep learning, activation functions are pivotal in determining a neural network’s output. For our research, we adopted the HardSwish activation function, which has showcased notable advantages over the conventional ReLU function. Mathematically, ReLU is expressed as in Equation (Equation 4).
(4)f(x)=max(0,x)

In contrast, HardSwish is described in Equation (Equation 5).
(5)$$f\left(x\right)=x\times \frac{\min(\max(0,x+3),6)}{6}$$

This distinction is visually represented in Figure 5, where the blue curve depicts the ReLU function that nullifies all negative values, and the green curve illustrates the HardSwish function, which offers a smoother gradient and does not nullify negative values as abruptly. Furthermore, a common challenge with ReLU is the ’dying neuron’ problem, where specific neurons become inactive and stop learning. With HardSwish, we improved the overall performance of our network model and achieved better results in our experiments. Our results, detailed in Section 6 and summarized in Table 3, indicate that the integration of the HardSwish activation function significantly enhances the performance of our model. This improvement suggests that HardSwish may be advantageous in a wide range of deep learning applications.

### 4.6. Evaluation Metrics

In HPE tasks, evaluation metrics are crucial for measuring model performance. Among the commonly used metrics, we adopted PCK (Percentage of Correct Keypoints) and Mean@0.1 for our evaluation. PCKh is a specific variant of PCK. Instead of using absolute distances, PCKh leverages the head bone link length as a reference. A prediction is deemed correct if the distance between the predicted and actual keypoints is within 50% of this reference length, denoted as PCKh@0.5. Meanwhile, Mean@0.1 quantifies the average discrepancy between predicted and actual keypoints, but the head bone link length normalizes it. This normalization makes it scale-invariant, ensuring consistent evaluation across different image resolutions and subject sizes.

## 5. Experimental Results and Discussion

Our comparative analysis involved an array of models, evaluated across distinct input resolutions of 256×256, 384×288, and 384×384, as illustrated in Table 1. The baseline models, which included SimpleBaseLine [6], PRTR [24], HRNet-W32 [25], and macro–micro [26] configurations, were examined at 256×256 and 384×384 for a subset of configurations.

When considering the baseline models at 256×256 and 384×384 resolutions, the SOCA-PRNet34 model demonstrated a remarkable PCKh@0.5 score of 89.846 at 256×256, which further increased to 90.877 at 384×384. These scores were significantly higher than those achieved by SimpleBaseline [6] and HRNet-W32 [25], as evidenced by the Mean@0.1 scores of 36.417 and 41.137, respectively, underlining the effectiveness of SOCA-PRNet’s approach. The model’s robustness to resolution scaling was particularly notable when compared to these benchmarks. The exceptional efficiency of SOCA-PRNet34 is further emphasized by its requirement of only 30 million parameters for training, which is considerably lower than the baseline models. This parameter efficiency, juxtaposed with its superior performance metrics, highlights the unique advantages of SOCA-PRNet over SimpleBaseline [6] and HRNet-W32 [25].

The SOCA-PRNet18 model was also evaluated across the same input size range. In line with the SOCA-PRNet34, the SOCA-PRNet18 surpassed baseline models in terms of PCKh@0.5 and Mean@0.1 metrics while operating with fewer parameters, reinforcing the efficacy of our approach. Furthermore, the HRNet-W32 [25] model demonstrated commendable performance with a PCKh@0.5 of 90.300 at the 256×256 resolution. However, even this strong competitor was marginally outperformed by our SOCA-PRNet models at higher resolutions, as reflected in the Mean@0.1 scores.

Figure 6 visually contrasts the accuracy and parameter counts of various 2D HPE models, including our SOCA-PRNet18 and SOCA-PRNet34, as well as the Pose-Resnet series of SimpleBaseline [6] and Pose-hrnet32 [25], which are listed in Table 2. This comparison highlights the efficiency and performance balance achieved by different models. Notably, SOCA-PRNet34 excels with a high PCKh@0.5 score of 90.875, using only 30 million parameters, demonstrating an optimal balance between accuracy and model economy. This is particularly impressive when compared to models like Pose-Resnet152, which has over double the parameters but similar accuracy levels. SOCA-PRNet18 also performs competitively, achieving close accuracy to more complex models with just 21 million parameters, illustrating the effectiveness of our approach in resource-limited scenarios. This analysis demonstrates the strength of SOCA-PRNet models in providing high accuracy with a reduced parameter count, affirming the success of our architectural optimizations in 2D HPE.

The data we present in graphical form offer a more intuitive understanding of our research outcomes. Specifically, Figure 7a displays the PCKh@0.5 scores for individual joints, contrasting the performance of our proposed SOCA-PRNet models against the SimpleBaseline [6] at a resolution of 256×256. Notably, SOCA-PRNet34 demonstrates significant improvements in challenging keypoints like the Elbow, Wrist, Hip, Knee, and Ankle over the PoseResNet baseline models. SOCA-PRNet18, despite being slightly less performant than its SOCA-PRNet34 counterpart, also shows a competitive edge, particularly in accurately estimating the Hip and Knee joints. These outcomes are significant as they highlight the efficacy of the SOCA-PRNet models, which are specifically designed to balance reduced model complexity with high accuracy. The superior performance of SOCA-PRNet34 in keypoints like the Hip and Knee, even surpassing the more complex PoseResNet152 model, underscores the success of integrating the SOCA module and our streamlined architecture. This balance is crucial for applications where model efficiency is as important as accuracy, particularly in real-world scenarios with limited computational resources.

In Figure 7b, the Mean and Mean@0.1 scores across all joints offer a clear comparison of our SOCA-PRNet models against the PoseResNet baseline models. The SOCA-PRNet34 stands out with the highest Mean accuracy of 90.877% and Mean@0.1 score of 41.137%, indicating its superior overall accuracy and precision in joint localization. This performance, especially in the Mean@0.1 metric, highlights its capability in accurately detecting joints in challenging conditions. The SOCA-PRNet18 also demonstrates notable performance, outperforming the PoseResNet50 and PoseResNet101 in Mean accuracy, which reinforces the effectiveness of our model design. Although slightly behind SOCA-PRNet34, it maintains high accuracy with fewer parameters. Comparatively, the PoseResNet152, despite its competitiveness, does not match the performance of SOCA-PRNet34, emphasizing the advancements our models bring in balancing efficiency and accuracy in 2D HPE.

The SOCA-PRNet18 model was also evaluated across the same input size range. In line with the SOCA-PRNet34, the SOCA-PRNet18 surpassed baseline models in terms of PCKh@0.5 and Mean@0.1 metrics while operating with fewer parameters, reinforcing the efficacy of our approach. Furthermore, the HRNet-W32 model demonstrated commendable performance with a PCKh@0.5 of 90.300 at the 256×256 resolution. However, even this strong competitor was marginally outperformed by our SOCA-PRNet models at higher resolutions, as reflected in the Mean@0.1 scores. This trend is visualized in Figure 6, which showes our models’ competitive edge in accuracy and model economy.

The data we present in graphical form offer a more intuitive understanding of our research outcomes. Specifically, Figure 7a displays the PCKh@0.5 scores for individual joints, contrasting the performance of our proposed SOCA-PRNet models against the baseline models at a resolution of 256×256. This visual comparison highlights the relative proficiency of each model in joint estimation accuracy. Complementing this, Figure 7b aggregates the performance metrics, presenting a concise overview of the Mean and Mean@0.1 scores across all joints at the same resolution. These collective metrics serve to encapsulate the models’ precision in joint localization in a single, comparative glance.

To contextualize our findings within practical applications, Figure 8 illustrates the practical efficacy of the SOCA-PRNet34 model by showing its pose estimation results on images from the MPII dataset. This visual representation demonstrates the model’s real-world applicability and solidifies its potential for accurate human pose estimation in varied and complex scenarios.

## 6. Ablation Study

Our ablation study aimed to optimize performance in 2D HPE with a focus on the individual contributions of different components that are deconvolution layers, the SOCA module, activation functions, and optimizers.

### 6.1. Initial Model Development

We initiated with the ResNet34 architecture, leading to our base model, PRNet34-3xDeconvolution. Equipped with three deconvolution layers, this model achieved a Mean accuracy of 86.607 and a Mean@0.1 value of 24.421, as presented in Table 3.

### 6.2. Assessing Deconvolution Layers

To assess the impact of additional deconvolution layers, we subsequently developed the PPNet34-5xDeconvolution model with five deconvolution layers. This variant showed an increase in Mean accuracy to 89.113 and Mean@0.1 to 37.359, in Table 3, indicating the effectiveness of added deconvolution layers in visual processing.

### 6.3. Evaluating the SOCA Module

Our exploration, however, was not limited to these results. Recognizing the potential of integrating global context into our model, we introduced GCN. This led to the emergence of the SOCA-PRNet34 model. The novel SOCA module, embedded within, integrates feature representations from varied layers. The resultant architecture thus becomes adept at capturing intricate details through layered representations while maintaining spatial awareness. The outcomes of this integration were obvious: SOCA-PRNet34 outperformed its predecessors, achieving an impressive Mean accuracy of 90.877 and a Mean@0.1 of 41.137, as presented in Table 3.

### 6.4. Comparative Analysis of Activation Functions and Optimizers

Further, we conducted a comparative analysis of activation functions and optimizers across these model variants. For PPNet34-5xDeconvolution, switching from ReLU and Adam to HardSwish and AdamW improved Mean accuracy from 88.925 to 89.113. In the case of SOCA-PRNet34, the ReLU-Adam configuration achieved a Mean accuracy of 88.259, while the AdamW optimizer, particularly when paired with HardSwish, elevated performance to a Mean accuracy of 90.877, as detailed in Table 4. This demonstrates the notable influence of activation functions and optimizers on model performance.

Our systematic ablation study reveals that each component—additional deconvolution layers, the SOCA module, and the choice of activation function and optimizer—independently contributes to enhancing the performance of 2D HPE models. These insights highlight the potential of component-specific optimizations in advancing the field.

## 7. Conclusions and Future Work

In this study, we introduced the SOCA-PRNet for 2D HPE, a novel approach that binds the efficient ResNet34 architecture to find a balance between computational simplicity and visual processing capability. The model’s design is further encouraged by including GCBs in the downsampler and upsampler modules, ensuring the assimilation of comprehensive global context features. Our proposed SOCA module plays a crucial role in merging and directing features with heightened spatial attention, allowing the model to generate detailed hierarchical representations. When compared to standard models on the MPII dataset, SOCA-PRNet’s enhanced performance becomes evident, driven by its refined feature processing, optimal activation function, and advanced optimizer. As we look to the future, SOCA-PRNet’s adaptability presents it as a promising option for applications beyond 2D HPE, such as 3D human pose estimation, object recognition, and hand pose estimation. Given its versatility, the model is anticipated to contribute significantly to enhancing interactive experiences in the rapidly expanding fields of HCI, robotics, and gaming.

## Figures and Tables

**Figure 1 sensors-24-00110-f001:**
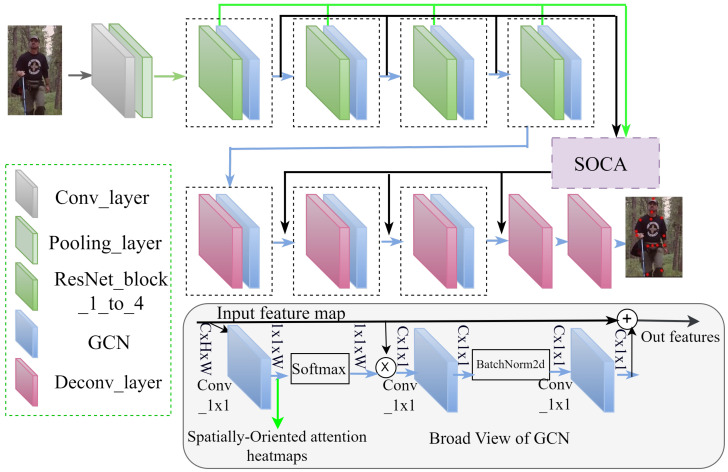
Detailed architecture of our proposed SOCA-PRNet model for 2D HPE.

**Figure 2 sensors-24-00110-f002:**
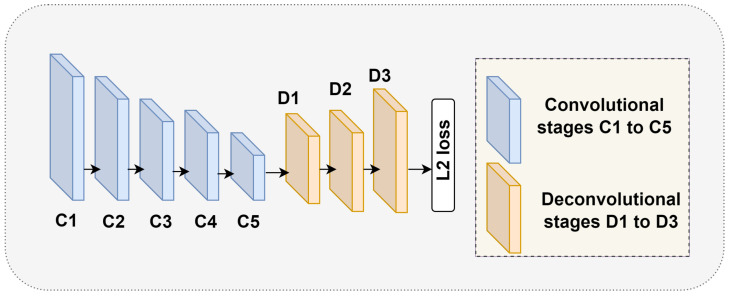
Detailed architecture of the simple base line model for 2D HPE [6].

**Figure 3 sensors-24-00110-f003:**
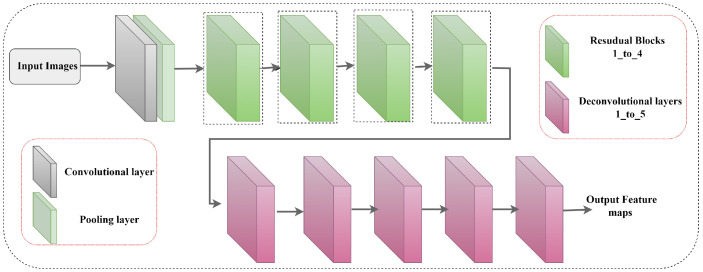
Modified ResNet with deconvolution module.

**Figure 4 sensors-24-00110-f004:**
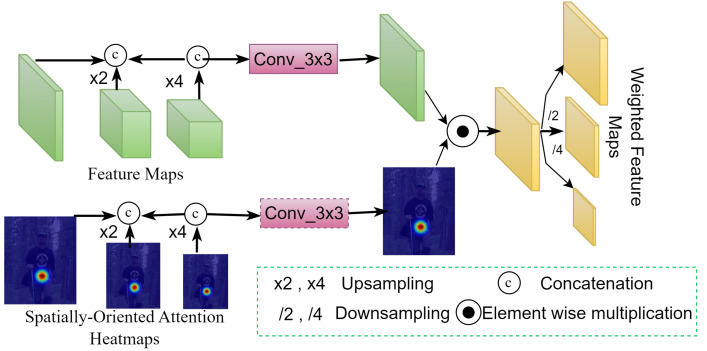
Visualization of the SOCA module’s feature integration and weights *W* generation and distribution mechanisms.

**Figure 5 sensors-24-00110-f005:**
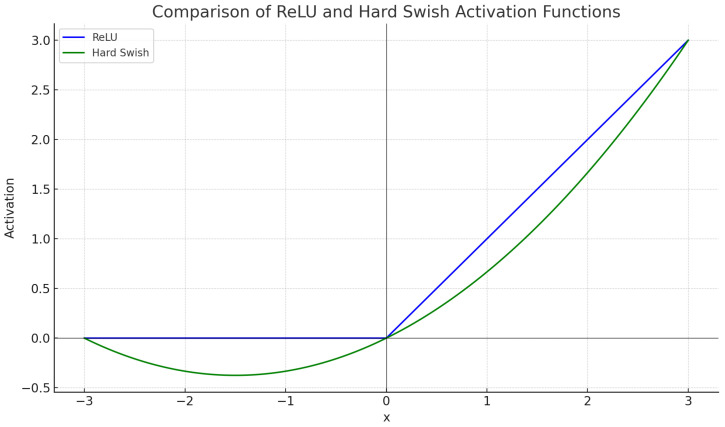
Comparative visualization of HardSwish and ReLU activation functions.

**Figure 6 sensors-24-00110-f006:**
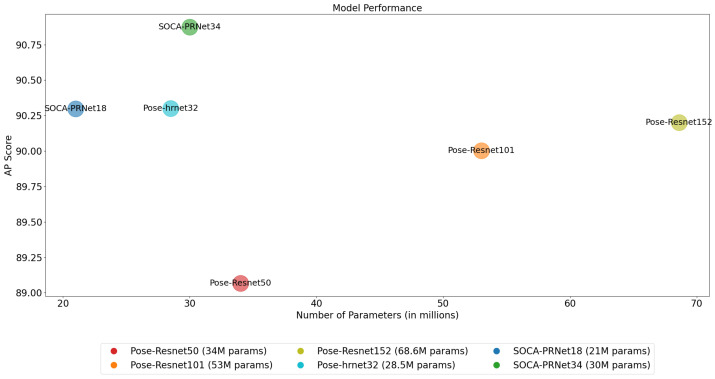
Visual analysis of 2D HPE models in terms of accuracy and parameter count.

**Figure 7 sensors-24-00110-f007:**
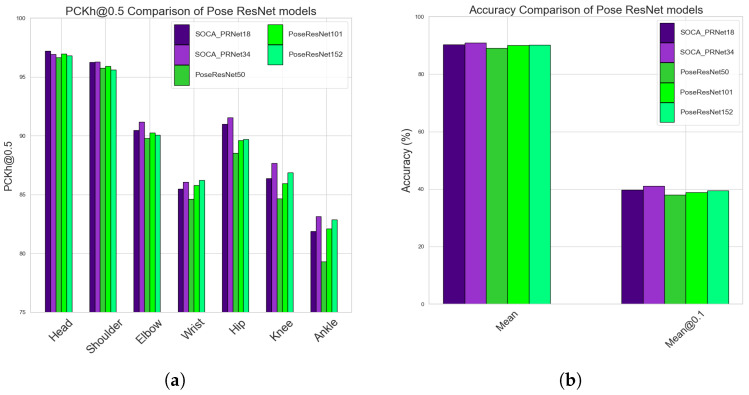
Graphical Illustration of the proposed model and simple baseline models. (**a**) Illustration of PCKh@0.5 results: proposed model and simple baseline models. (**b**) Graphical analysis of Mean and Mean@0.1: proposed models and simple baseline models.

**Figure 8 sensors-24-00110-f008:**
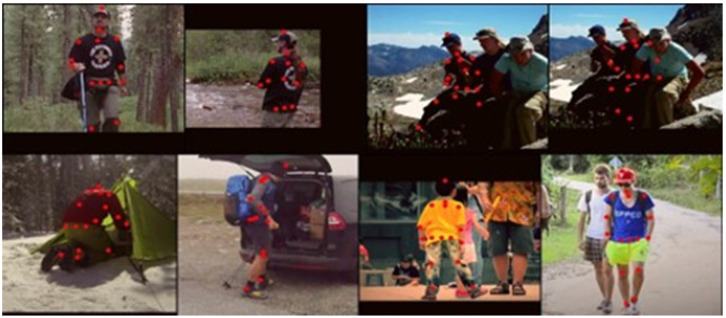
Qualitative results on MPII pose estimation result, containing viewpoint change, occlusion, and self-occlusion.

**Table 1 sensors-24-00110-t001:** Performance comparisons of our SOCA-PRNet with other model results on MPII dataset.

Model	Backbone	Input	Head	Sh	Elbow	Wrist	Hip	Knee	Ankle	Mean	Mean@0.1
SimpleBaseLine [6]	Pose-Resnet50	256×256	96.351	95.329	88.989	83.176	88.420	83.960	79.594	88.532	33.911
		384×384	96.658	95.754	89.790	84.614	88.523	84.666	79.287	89.066	38.046
	Pose-Resnet101	256×256	96.862	95.873	89.518	84.376	88.437	84.486	80.703	89.131	34.020
		384×384	96.965	95.907	90.268	85.780	89.597	85.935	82.098	90.003	38.860
	Pose-Resnet152	256×256	97.033	95.941	90.046	84.976	89.164	85.311	81.271	89.620	35.025
		384×384	96.794	95.618	90.080	86.225	89.700	86.862	82.853	90.200	39.433
PRTR [24]	Pose-Resnet101	384×384	96.300	95.000	88.300	82.400	88.100	83.600	77.400	87.900	-
	Pose-Resnet152		96.400	94.900	88.400	82.600	88.600	84.100	78.400	88.200	-
HRNet-W32 [25]	Pose-hrnet_w32	256×256	97.100	95.900	90.300	86.400	89.100	87.100	83.300	90.300	37.700
macro–micro [26]	Pose-Resnet50	256×256	96.650	95.490	89.220	83.650	88.290	84.440	80.910	88.890	-
SOCA-PRNet	Pose-Resnet18	256×256	96.965	95.688	89.398	84.051	90.254	85.029	80.728	89.425	34.483
		384×288	97.169	95.788	90.131	84.462	90.341	85.331	81.696	89.766	36.435
		384×384	97.203	96.264	90.472	85.489	90.981	86.379	81.890	90.297	39.670
	Pose-Resnet34	256×256	97.237	95.805	90.012	84.891	90.064	85.976	81.507	89.846	36.417
		384×288	97.271	96.247	90.608	85.642	91.016	86.984	82.712	90.536	38.158
		384×384	96.930	96.298	91.188	86.072	91.535	87.668	83.137	90.877	41.137

**Table 2 sensors-24-00110-t002:** Comparison of model complexity: parameter counts in various models.

Model	No. Parameters (M)
Pose-Resnet50 [6]	34.0
Pose-Resnet101 [6]	53.0
Pose-Resnet152 [6]	68.6
Pose-hrnet32 [25]	28.5
SOCA-PRNet 18	21.0
SOCA-PRNet 34	30.0

**Table 3 sensors-24-00110-t003:** An ablation analysis: evaluating network performance across varied additional modules.

Model	Input	Head	Shoulder	Elbow	Wrist	Hip	Knee	Ankle	Mean	Mean@0.1
PRNet34-3 × Decon	384 × 384	95.805	94.463	86.978	80.400	87.467	80.940	75.200	86.607	24.421
PPNet34-5 × Decon		96.351	95.312	89.671	84.616	88.523	84.807	80.563	89.113	37.359
SOCA-PRNet34		96.930	96.298	91.188	86.072	91.535	87.668	83.137	90.877	41.137

**Table 4 sensors-24-00110-t004:** Comprehensive Ablation analysis: assessing network performance across distinct activation functions and optimization techniques.

Model	Activation Function	Optimization	Mean	Mean@0.1
PPNet34-5 × Decon	ReLU	Adam	88.925	36.784
	HardSwish	AdamW	89.113	37.359
SOCA-PRNet34	ReLU	Adam	88.259	38.457
	ReLU	AdamW	89.415	39.493
	HardSwish	AdamW	90.877	41.137

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
