# Peer review of "SOCA-PRNet: Spatially Oriented Attention-Infused Structured-Feature-Enabled PoseResNet for 2D Human Pose Estimation"

_sensors, 2023, doi:10.3390/s24010110_

Round 1
Reviewer 1 Report
Comments and Suggestions for Authors
This paper presents a novel architecture called SOCA-PRNet for 2D human pose estimation. The proposed model builds upon a ResNet34 backbone and incorporates several enhancements including Global Context Blocks, deconvolution layers, and a novel Spatially-Oriented Attention module. The contributions are clearly stated and the overall methodology is sound. However, there are some major issues that need to be addressed before the paper can be accepted for publication:
- I appreciate the novel contributions proposed in this work, including the SOCA module and incorporation of Global Context Blocks into the ResNet backbone. However, more rigorous experimentation and comparisons are required to demonstrate the effectiveness of the complete framework. Addressing the above major concerns, as well as the minor issues, will significantly improve the quality and clarity of the manuscript. I hope these comments are constructive in strengthening the paper. With suitable revisions, this work could represent a valuable advancement for the field of 2D human pose estimation.
- The ablation studies evaluating the impact of different components like SOCA, activation functions, and optimizers are insufficient. Each modification is added sequentially on top of the others, so it is unclear how much each one individually contributes. The authors should conduct a more systematic ablation where the presence/absence of each component is independently varied.
- The superiority of SOCA over traditional skip connections for combining multi-scale features has not been conclusively demonstrated. Additional experiments/analysis are needed to show its benefits over existing approaches.
- There are no comparisons to other state-of-the-art 2D pose estimation methods like HRNet, SimpleBaseline, CPN, etc. Such comparisons are essential to contextualize the performance of SOCA-PRNet.
- The number of model parameters for SOCA-PRNet and other reference models should be reported. This provides insight into efficiency.
- The authors claim the model is efficient but do not provide any benchmarking of runtime, FPS, etc. to quantify this. Such implementation analysis is needed.
- The MPII dataset used is quite old. Evaluation on more recent and challenging benchmarks like COCO would strengthen the results.
- The paper needs better organization and flow. The methodology section jumps straight into architectural details without any high-level overview. A clear, step-by-step description of the model components would improve readability.
- The introduction could provide more background on the evolution of deep learning for 2D pose estimation to motivate the proposed approach.
- Figures 4 and 5 are not referenced in the text. The content/purpose of each figure should be clearly explained.
- The English needs proofreading for grammatical errors throughout.
This paper presents a novel architecture called SOCA-PRNet for 2D human pose estimation. The proposed model builds upon a ResNet34 backbone and incorporates several enhancements including Global Context Blocks, deconvolution layers, and a novel Spatially-Oriented Attention module. The contributions are clearly stated and the overall methodology is sound. However, there are some major issues that need to be addressed before the paper can be accepted for publication:
- I appreciate the novel contributions proposed in this work, including the SOCA module and incorporation of Global Context Blocks into the ResNet backbone. However, more rigorous experimentation and comparisons are required to demonstrate the effectiveness of the complete framework. Addressing the above major concerns, as well as the minor issues, will significantly improve the quality and clarity of the manuscript. I hope these comments are constructive in strengthening the paper. With suitable revisions, this work could represent a valuable advancement for the field of 2D human pose estimation.
- The ablation studies evaluating the impact of different components like SOCA, activation functions, and optimizers are insufficient. Each modification is added sequentially on top of the others, so it is unclear how much each one individually contributes. The authors should conduct a more systematic ablation where the presence/absence of each component is independently varied.
- The superiority of SOCA over traditional skip connections for combining multi-scale features has not been conclusively demonstrated. Additional experiments/analysis are needed to show its benefits over existing approaches.
- There are no comparisons to other state-of-the-art 2D pose estimation methods like HRNet, SimpleBaseline, CPN, etc. Such comparisons are essential to contextualize the performance of SOCA-PRNet.
- The number of model parameters for SOCA-PRNet and other reference models should be reported. This provides insight into efficiency.
- The authors claim the model is efficient but do not provide any benchmarking of runtime, FPS, etc. to quantify this. Such implementation analysis is needed.
- The MPII dataset used is quite old. Evaluation on more recent and challenging benchmarks like COCO would strengthen the results.
- The paper needs better organization and flow. The methodology section jumps straight into architectural details without any high-level overview. A clear, step-by-step description of the model components would improve readability.
- The introduction could provide more background on the evolution of deep learning for 2D pose estimation to motivate the proposed approach.
- Figures 4 and 5 are not referenced in the text. The content/purpose of each figure should be clearly explained.
- The English needs proofreading for grammatical errors throughout.
Author Response
We are grateful for the constructive feedback provided by the reviewer. We have meticulously reviewed each comment and have made comprehensive revisions to our manuscript in response. Our efforts aim to address the concerns raised and enhance the overall quality of our work. Once again, thank you so much.

Reviewer 2 Report
Comments and Suggestions for Authors
This paper presents a deep neural network architecture for two dimensional human pose estimation. However the proposal needs to be clearly presented in section 3.
1)Algorithm 1 does not represent any specific. It is too general to be included.
2) Figure 1 should represent clearly the distinction between the existing networks and the proposed network.
3) In section 3.1, the line " We modified .....convolution and a pooling layer" should be illustrated by a figure, clearly showing the modification.
4) The whole working process should be explained with figures sequentially clearly making distinctions from the existing methods. In other words authors should rewrite section 3 focusing their contribution over existing algorithms. Now it seems like a combination of different existing works.
Minor point:
At the last line of section 3.3, table no is not compiled.
Author Response

(The authors gave the same response as above.)

Round 2
Reviewer 1 Report
Comments and Suggestions for Authors
The authors have revised the manuscript according to the reviewer comments. The present form the paper can be accepted for the publication.